# FEM Numerical and Experimental Work on Extrusion Welding of 7021 Aluminum Alloy

**DOI:** 10.3390/ma16175817

**Published:** 2023-08-24

**Authors:** Dariusz Leśniak, Wojciech Libura, Beata Leszczyńska-Madej, Marek Bogusz, Jacek Madura, Bartłomiej Płonka, Sonia Boczkal, Henryk Jurczak

**Affiliations:** 1Faculty of Non-Ferrous Metals, AGH University of Science and Technology, 30-059 Krakow, Poland; libura@agh.edu.pl (W.L.); bleszcz@agh.edu.pl (B.L.-M.); bogusz@agh.edu.pl (M.B.); madura@agh.edu.pl (J.M.); 2Łukasiewicz Research Network—Institute of Non-Ferrous Metals, Light Metals Division, 44-100 Gliwice, Poland; bartlomiej.plonka@imn.lukasiewicz.gov.pl (B.P.); sonia.boczkal@imn.lukasiewicz.gov.pl (S.B.); 3Albatros Aluminum Corporation, 61-102 Poznan, Poland; h.jurczak@albatros-aluminum.com

**Keywords:** AlZnMg alloys, extrusion welding, porthole die geometry, welding conditions, seam weld quality

## Abstract

Extrusion welding of AlZnMg alloys encounters great technological difficulties in practice associated with high shaping forces and the low quality of longitudinal welds. Three different chemical compositions of 7021 aluminum alloy, differing in terms of Zn and Mg contents, were used in the first stage of the research. The laboratory device modelling the behavior of metal in welding chambers of the porthole die was applied to examine the ability of 7021 alloys to produce high-quality joints. The weldability tests were carried out for different welding temperatures—400, 450 and 500 °C—and for a fixed welding pressure of 300 MPa. The microstructural effects in pressure-welds were evaluated with the use of OM and SEM/EDS. The temperature–pressure parameters in the welding chambers were analyzed by using the FEM method for original porthole dies while extruding tubes with dimensions of Ø50 × 2 mm. Finally, the industrial extrusion trials were performed with examination of the structure and strength of the seam welds. It was found that it is possible to produce high-quality high-strength welds in tubes extruded from AlZnMg alloys in industrial conditions (the strength of welds in the range of 96–101% of the strength of the basic non-welded material) through properly matched alloy chemical composition of the alloy, construction of the porthole dies and temperature–speed conditions of deformation.

## 1. Introduction

The hot extrusion of profiles with the use of porthole dies is a common production technique. In this process, which is also called welding extrusion, the heated billet is divided into inlet channels of the die, and then the separate streams of metal enter the welding chamber, and they finally flow through the die cavity in the form of a hollow section. The obtained profile contains the longitudinal welds, the number of which depends on the die construction. The specific microstructure of the welds may be the reason for the poor strength of the profile.

The process is traditionally applied to extrude aluminum alloys, e.g., the 6xxx series. It is known from industrial practice that extrusion welding of high-strength alloys such as AlCuMg, AlMg3–5 (Mg3–5 in aluminum alloy represents the Mg chemical composition in mass percentage), AlZnMg and AlZnMg (Cu) causes serious difficulties in terms poor weldability and high deformation resistance. The process parameters, such as temperature and pressure in the welding chamber, extrusion speed and particularly porthole die design play an important role in effective welding. 

Khan et al. [1] observed the critical role of welding chamber geometry on weld quality in 6082 and 7008 alloys. Formation of voids containing oxygen under the bridge is responsible for the poor strength of the welds. Similar observations were revealed by Oosterkamp [2]. Duplancic [3] investigated the quality of welds in relation to the porthole die geometry and process parameters. For the AlZnMg1 alloy to guarantee the good strength of the welds, the normal pressure in the welding chamber should be 6-fold higher than the flow stress of the alloy, whereas this factor should be 10-fold greater for AlZnMgCu1.5.

Donati and Tomesani [4] studied weld quality in connection with the porthole die geometry and process parameters for 6060 and 6082 alloys. The inlet channels should be wide for good weld quality and high exit speed. They proposed a new kind of die, a “butterfly” type, which has been implemented in industrial practice and increased the exit speed by 20%.

The results of FEM simulations of porthole die extrusion are presented in many works [5,6,7,8,9]. The main research stream therein is focused on the distribution of hydrostatic pressure and temperature in a welding chamber of the die. The data presented indicate that the pressure value in the welding chamber is responsible for the good quality of the welds. The pressure in turn is determined by the geometry of the inlet channels, the dimensions of the welding chamber, the length of the bearing land, the temperature and extrusion speed as well as the deformation value in the process. Akeret [10] proposed a qualitative criterion for welding the materials in the extrusion process. For the first time, he described the weldability as the relationship between the normal pressure *σ_n_* in a welding area and the yield stress *k* of the metal. He provided limited values for different AlMgSi alloys leading to successful welding. Plata and Piwnik [11] presented an energetic method for longitudinal extrusion welding in which the contact surface, the contact time and the normal pressure to the yield stress ratio play the most important roles. A similar criterion presented by Donati and Tomesani [12] introduced a factor correcting the real time of contact.

Theoretical criteria for extrusion welding can be used in process optimization in connection with a numerical simulation of the process which allows predicting fundamental process parameters and conditions for welding. Ceretti [13] investigated the criterion of Plata and Piwnik based on the simulation of the extrusion of the 6061 alloy. 

Many authors performed studies on the weldability of alloys by using hot working processes, e.g., upsetting, rolling or hot extrusion [14]. However, there are no tests that fully control the welding conditions that occur in extrusion including the fact that the welding process is run without contact with air atmosphere.

The effects of different geometries of the weld chamber and the processing conditions on the quality of the weld seam are investigated by Baker [15]. Through computer simulations, conditions related to weld seam formation were modelled and compared with the experimental results. The experimental results demonstrate that metal flow controlled by the die geometry causes defects, leading to the inferior mechanical performance of the extrudate. A comprehensive microstructural characterization of welds is presented. The authors of the study [16] introduced the analysis of different quality criteria of longitudinal seam welds that appear in aluminum profiles during extrusion using porthole dies. A new improved dimensionless welding quality criterion is proposed by the authors. The quantitative correspondence between welding types and the values of the developed criterion is also defined. In the study [17], porthole extrusion using different depths of welding chambers were designed and manufactured. Profiles extruded with different depths of the welding chambers were obtained by performing extrusion experiments. The welding quality of the extruded profiles was characterized by microstructure observation, a tensile test and fracture analysis. In the study [18], the welding quality of a 6063 aluminum alloy hollow square tube extruded by a porthole die was studied. The K criterion was introduced to evaluate the welding quality. The DRX fraction, grain size, and its influence on the welding quality were analyzed. They found that the uniform microstructure near the welding line would also affect the welding quality. Porthole die extrusion of Mg-Al-Zn alloy was conducted using the as-cast, as-homogenized and as-extruded billets [19]. The effects of the initial microstructure on the texture and mechanical properties of the extruded profiles were investigated. The results showed that complete dynamic recrystallization (DRX) occurred in the welding zone of all profiles, and the profile extruded from the as-cast billet had the smallest DRXed grains. Fine nanostructures of the bonding interfaces of weld seams formed by porthole die extrusion in the absence/presence of a gas pocket behind the bridge of the extrusion die were studied in [20] to understand interfacial bonding mechanisms. It was found that the formation of adverse gas pockets can be avoided by increasing the depth of the welding chamber. 

An experimental setup is proposed [21] in order to analyze the welding condition of aluminum specimens under different loading conditions and temperatures. Based on backward cup extrusion, two specimens were compressed together within a heated container. Further, microstructural analysis of the welded specimens was conducted to observe if solid-state bonding occurred. The results of the trials are used to analyze the quality criteria for the seam welds during extrusion processes.

In the research [22], the commercial FEM software package, DEFORM v13.1, was used to simulate experiments in order to understand the relationship between the bridge design and the thermal mechanical history. The bridge can be divided into two parts: the lower part, close to the welding chamber, and the upper part, which initially split the billet into metal streams. The results showed that the geometry of the lower bridge influenced extrusion loads and profile exit temperatures. In contrast, changes to the geometry of the upper bridge had little effect on the porthole die extrusion process.

Different types of welding were found during porthole die extrusion by varying the extrusion die structure and process parameters [23]. It was found that the welding quality is determined by metal flow behavior, the solid-state bonding process and microstructural evolution. The formation of the unbonded macrodefects in the extruded profile is affected by improper metal flow behavior, while the formation of the microvoids on the bonding interface is caused by insufficient solid-state bonding. The influence of billet heating temperature and extrusion speed on the microstructure and mechanical properties of welding seams was studied in [24]. It was found that, in porthole die extrusion of aluminum alloy profiles, fine or coarse grains and microvoids can be formed in welding seams. The hardness, strength and ductility of the extruded profiles can be improved by increasing billet heating temperature and extrusion speed.

The metal flow law in the unsteady stage during the porthole die extrusion process was studied by Wang et al. [25]. They proposed the interfacial bonding mechanism of the longitudinal welds in the unsteady zones of the profiles and a method for evaluating the length of the unsteady zones of the profiles. The study by Annadurai et al. on extrusion die design using FEM simulations allowed for a reduction in experimental trials by 2/3 and a doubling of the lifespan of the die [26]. The die design and FEM simulations show the potential to predict extrusion defects and final profile microstructure, texture and mechanical properties [27,28]. Yu et al. used FEM model predictions to understand the effect of the bridge shape on the textures that formed along the weld seam during porthole die extrusion [29].

In the study [30], the dynamic recrystallization (DRX) behavior of an Al-Zn-Mg alloy during the porthole die extrusion process was studied. Higher deformation temperature and lower strain rate are favorable for the occurrence of DRX. The volume fraction of DRX at the zones close to bridge and porthole wall is much higher than that in the other zones. The objective of the study [31] was to develop a new porthole extrusion die for improving the welding pressure in the welding chamber by using numerical analysis. Through numerical analysis, the welding pressures in the welding chamber between the new porthole die and the conventional porthole die were compared with each other. The shape of the porthole die bridges was studied by FEM [32]. They found that dies that have curved bridges offer optimum process conditions. A Taguchi method and ANOVA were used to optimize the geometry of the welding chamber and other process parameters [33]. An optimal combination of the investigated parameters was determined for all the process outputs according to specific process needs.

The effect of process parameters on longitudinal welding quality was studied by Chen et al. [34]. The morphology of the poor longitudinal weld seam is observed by optical microscopy and scanning electron microscopy. Simulation results show that a lower extrusion speed, higher billet temperature and larger billet diameter are beneficial to improve longitudinal welding quality. To improve the flow balance in the die, a design approach was introduced to find an appropriate die structure that includes the porthole and pocket geometry correction, the bearing length adjustment, and the bridge structure modification [35]. Using the proposed die, the predicted velocity relative difference (VRD) and the maximum velocity difference (ΔV) of the extrudate were significantly lower than those of an initial die.

This work presents an innovative way for predicting seam weld quality in a tube of Ø50 × 2 mm from 7021 alloy. The method employs a patented laboratory device to investigate the welding process, allowing full modelling of conditions in a welding chamber of the porthole die [36]. This method allows reducing the time-consuming and costly industrial extrusion trials. Generally, the AlZnMg and AlZnMgCu alloys are known as difficult to weld during extrusion through porthole dies, so the proper choice of the alloy composition and the proper design of the porthole die construction are very challenging for these aluminum alloys. The FEM simulations as well as the industrial verification complete the research procedure. In the first stage, the Akeret indicator $\frac{\sigma_{n}}{k}$ was determined in relation to the chemical composition of the alloy. The Akeret indicator σ_n_/k is a welding parameter for the susceptibility of the material to extrusion welding. The lower the value of this indicator, the higher the susceptibility to extrusion welding. For example, for easily deformable and easily weldable AlMgSi alloys, the value of this parameter is 2–3, while for difficult-to-deform alloys with lower weldability, this parameter is in the range of 6–10 (σ_n_ is the normal stress affecting the weld in the welding chamber of porthole die; k is the flow stress of the deformed material). The second stage comprised FEM simulations and provided information where the required value of the indicator above was obtained in the welding chambers of different geometries. The third stage comprised microstructural analysis of the welds and mechanical testing of samples including welds; and for comparison, these were taken from the base material without welds.

## 2. Materials and Methods

### 2.1. Characterization of AlZnMg Alloys

The billets, with the chemical composition presented in Table 1 and a diameter of 178 mm, were DC cast in semi-industrial conditions. Three alloys were investigated at a EN AW-7021 grade. In the case of all examined alloys, the low-melting microstructure components were dissolved during homogenization by soaking to a degree sufficient in practice—no incipient melting peaks on the DSC curves are noted. As a result, the significant increase in solidus temperature was achieved, and the obtained values are within the range from 559.2 °C for alloy 3 to 613.2 °C for alloy 1 (Table 2).

### 2.2. Method and Device for Testing Ability to Extrusion Welding

A new method and device for weldability testing was carried out using a developed authorial patented device (Figure 1a,b, [1]). This device enables repeating conditions of joining of a metal which are expected to exist in a welding chamber of the porthole die during extrusion of hollow sections (processes of shearing and compression without the air admission). The test of weldability consists of cutting of two samples from the tested alloy (separated by steel counter-samples) and then axial pressing (welding) at the assumed temperature and under the assumed unitary pressure (hydrostatic pressure). The cutting of the samples heated in the heating chamber (1) and placed in the shearing and welding cartridge tool (2) is conducted by the upper shearing punch (6). The axial pressure is conducted by the hydraulically driven compression stem (3), which affects the welding surface of the samples perpendicularly.

Figure 2 presents the subsequent stages of work of the device for weldability tests: a—the tool cassette is inserted into the heating chamber and the presser presses the sample packet; b—the samples are sheared with the upper punch while the presser is still held; c—after cutting, the shear punch stops and the compression punch hydraulically applies the specific pressure force on the welded samples. In this way, the situation is similar to that of the real porthole die, where separate streams of metal move on the channel surface and are welded in the welding chamber. Thus, the steel plate “plays role” of the 1 inlet channel.

Figure 3 shows the arrangement of the samples for the alloy tested (the orange and green ones) and steel counter-samples (the blue light and blue dark ones) in the tool cassette, in the starting position (left view) and in the end position (right view). The initial sample dimensions were of 60 × 10 × 10 mm. Dimensions of samples and counter-samples in the starting position are shown in Figure 4. 

### 2.3. Methodology of the Microstructural and Mechanical Examination

The method of taking samples from the extruded tube for further examination of the microstructure and mechanical properties is presented in Figure 5. Welding extrusion is the process of extruding hollow profiles using a bridge–porthole die. In this process, the metal from the input material (ingot) is cut on the die bridge, then flows through the inlet channels, successively flows into the welding chambers located under the bridge and finally flows out through the working opening of the die. The finished product has longitudinal welds in the cross-section (seam welds) located along the entire profile. The samples for microscopic examination were mounted in resin, mechanically ground with sandpaper with appropriate gradation, and then mechanically polished in two stages using a diamond paste suspension and a colloidal silicon oxide suspension for finishing polishing. To reveal the microstructure of the samples for observation with a light microscope, the samples were anodized in Barker reagent with the composition of 100 mL H_2_O + 2 mL HBF_4_. The microstructure of the samples was examined by means of light microscopy (OLYMPUS GX51 microscope, Tokyo, Japan) and scanning electron microscopy (Hitachi SU 70 microscope, Tokyo, Japan). Additionally, the chemical composition in the microareas was analyzed using the energy-dispersive X-ray spectroscopy (EDS) method (Thermo Fisher Scientific, Waltham, MA, USA). An analysis of the chemical composition within the grains was performed to determine the content of individual alloying elements. In each case, a minimum of 20 spot analyses were performed. The test was carried out at an acceleration voltage of 15 kV. In addition, the average grain diameter was determined in the weld area and outside of this area (average chord method), and the average grain elongation coefficient in the weld area was determined. Usually, the samples with a weld in the center are prepared for static tensile tests, allowing for an assessment of the quality of the joint by comparing the mechanical properties of samples with the weld and reference samples (without weld). Studies of the basic mechanical properties—yield strength (YS), ultimate tensile strength (UTS) and percentage elongation (A%)—were carried out using the INSPECT 100 strength machine (with a maximum tensile strength of 100 kN). 

Analysis of the crystallographic orientation of grains was performed under a high-resolution INSPECT F50 FEI scanning electron microscope with attachments for chemical analysis by EDS and a camera for EBSD. The EBSD analysis was performed using EDAX APEX^TM^ EBSD software. The cross-section samples were cut, ground and polished mechanically. In the last stage of preparation, the samples were polished using the Leica RES101 Ion Milling System. Maps were scanned at a resolution of 800 × 800 mm and a step of 1 mm.

### 2.4. FEM Numerical Modeling of Extrusion Welding

The extrusion process of tubes of Ø50 × 2 mm from aluminum 7021 alloy 2 was FEM modeled by using porthole dies of various geometries: conventional porthole die for 6xxx alloys based on the local 3-armed bridges (die 1) and die 2 with maximal broad inlet channels, shaped pockets, bearings of varied length and proper geometry of the central baffle and mandrels. Moreover, the thickness of the bridges for die 2 was increased by 4 mm, whereas their length was increased by 40 mm in relation to die 1. The dimensions of the porthole dies discussed are shown in Figure 6.

The overall dimensions for both dies are the same. The diameter of the dies is Ø310 mm and the thickness of the tool set assembly is 210 mm. In the case of die 1, the bridge part is 65 mm, the length of the bridges is 55 mm, and the thickness is 16 mm; the height of the welding chambers is 20 mm. The die plate with the backer that completes the set is 145 mm thick.

In the case of die 2, the bridge part is 103 mm, longer than that of die 1 by 38 mm. Longer bridges with a length of 95 mm and a thickness of 20 mm were used. The height of the welding chambers of the complex set is also greater, 25 mm. The die plate is 107 mm thick; and in this case, no backer is used. A chamfered contact surface was used at an angle of 20° instead of the parallel connection of the porthole and die plate as in die 1, providing stiffness and support without the use of a backer.

Conducting numerical simulations required a number of preparatory works to be performed, e.g., development of a digital three-dimensional CAD model of both the die and the entire toolkit enabling installation in an industrial press (Figure 7). Three-dimensional tool models were developed using SolidWorks Premium v2019 SP0.0 CAD/CAM software. Preparatory work and numerical simulations were carried out using QForm Extrusion v10.1.7 3D software. A special calculation module enables importing a CAD model of tools and generating a mesh of finite elements on their surface and in the volume of the model. The numerical simulation module consists of a pre-processor and a post-processor that provides visualization of the results [1]. The numerical model of the finite element simulations is defined based on the flow formula proposed by Zienkiewicz and Pittman, where the deformed material is treated as an incompressible and rigid viscoplastic continuum, while elastic deformations are neglected [2]. The calculations are based on the Euler–Lagrange model, which uses finite elements to simultaneously relate the material flow with the deformation and temperature distribution of the tool. This means that the elastic deformation of the die affects the way the metal flows, while the deformation of the tool itself is determined by the pressure of the metal on its surface. The software is based on two discrete models. The first of them, the Lagrange model, is designed to simulate the transient state of the initial stage of the process when the metal fills the die, while the second, the combined Euler–Lagrange model, is used to simulate the steady-state phase [3]. 

The mesh of finite elements, both surface and volume die set and workpiece, is made of elements with the geometry of triangles of various sizes connected by nodes. The mesh of the computational domain-aluminum filling the interior of the die as well as the die set was created automatically after defining the appropriate geometrical elements of the tool. The size of the local elements was selected using individually selected factors for each of the die areas. The default size factor is 1, the coefficient of the adaptation of the container is 1.0, the coefficient of the adaptation of the welding chamber is 0.95, the coefficient of the adaptation of the pocket and profile geometry is 0.45, and the coefficient of the adaptation of bearing is 0.80.

In the case of die 1, the number of nodes on the surface and in the volume of the workpiece is 220,146 and 261,203 for the die set—a total of 481,355 nodes. The number of mesh elements on the surface and in the volume of the workpiece is 1,149,535 and 1,392,619 for the die set—a total of 2,542,154 finite elements. In the case of die 2, the number of nodes on the surface and in the volume of workpiece is 225,440 (due to larger inlet channels and the volume of metal filling the die) and 232,893 for the die set—a total of 458,333 nodes. The number of mesh elements on the surface and in the volume of the workpiece is 1,173,504 and 1,243,138 for the die set—a total of 2,416,642 finite elements.

FEM mesh verification was carried out in a special QShape module for mesh preparation. The prepared surface and volume mesh are checked by the algorithm used in QShape for further reconstruction and potential errors—starting numerical calculations in the main calculation solver is not possible for an unverified mesh.

Obtaining high-accuracy simulation results requires a detailed definition of the rheological properties of the deformed material. The Hensel–Spittel constitutive Equation (1) was used to describe the deformation characteristics of the alloy:(1)$$\sigma_{p}=A\cdot e^{m_{1}T}\cdot T^{m_{9}}\cdot \varepsilon^{m_{2}}\cdot e^{\frac{m_{4}}{\varepsilon }}\cdot {\left(1+\varepsilon \right)}^{m_{5}T}\cdot e^{m_{7}\varepsilon }\cdot {\dot{\varepsilon }}^{m_{3}}\cdot {\dot{\varepsilon }}^{m_{8}T}$$
where 

σ—plastic stress,

ε—plastic strain,

$\dot{\varepsilon }$—strain rate, and

T—temperature of deformation. 

The concept of describing the yield stress with one curve makes it possible to determine empirically the rheological properties of the material in laboratory tests, e.g., in the high-temperature compression test. The effect of temperature during plastic deformation on the decrease in yield stress value should be taken into account. For this reason, the determination of the characteristics of the selected material using a wide range of temperatures and strain rates is crucial for the description of rheological properties used in material extrusion tests [4]. The coefficients from Equation (1) are shown in Table 3.

In the simulations, the friction model defined by Levanov (2) was adopted on the contact surface of the deformed metal and the tool.
(2)$$F_{t}=m\frac{\bar{\sigma }}{\sqrt{3}}\left(1-exp\left(-1.25\frac{\sigma_{n}}{\bar{\sigma }}\right)\right)$$
where m is the coefficient of friction, *σ_n_* is the perpendicular contact pressure and σ is the actual stress. The equation can be considered as a combination of the constant friction model and the Coulomb friction model. The second term in parentheses of Equation (2) takes into account the effect of normal pressure on surface contact. For high pressure values, the expression approximates the conditions defined by the constant friction model. On the other hand, for low pressure values, it defines a linear approximation depending on the normal stress at the surface contact. All the defined extrusion process parameters are presented in Table 4. 

### 2.5. Extrusion Trials of Round Tubes

The extrusion trials of the tube of Ø50 × 2 mm from 7021 alloy 2 by using the double-hole porthole die 1 and porthole die 2 (Figure 8) were carried out on the 7-inch hydraulic press of 25 MN capacity. The billet dimensions and the conditions of the trials were identical with these in the numerical simulation. Die 1 was based on the variant usually used during extrusion of the 6000 series alloys, whereas die No 2 was the modified version elaborated in FEM simulation of the investigated process. Figure 8 shows the layout of the inlet channels in both the tested dies. Despite the similarity in the inlet channel layout, the channels in die 2 are wider while the bridges are higher and thicker to withstand the increased extrusion force and to avoid elastic deflection of the die. The height of the welding chamber is higher in die 2. The die insert was equipped with pockets, which regulate metal flow in the region close to the die opening. In such a case, the metal flow is more uniform and this improves the geometrical stability and dimensional accuracy of the extruded profile. Alloy 2 was used in the experiments. The extruded tubes were air cooled on the run-out table and next submitted to ageing. During the extrusion trials, the ram speed, exit speed, extrusion pressure and profile temperature were recorded. The cracking on the extrudate surface means that the maximal extrusion speed was exceeded. The maximum exit speed during extrusion of the 7000 series alloys is usually very low (below 1.5 m/min), so each achievement in this field considerably improves process efficiency.

## 3. Results

### 3.1. Weldability Tests

Figure 9 shows the stress–strain curves recorded during static tensile testing of samples welded from alloys 1, 2 and 3 at 450 °C and 500 °C. The influence of the welding temperature on the mechanical properties of the welds produced in the weldability tests at a given unit pressure *p* = 300 MPa for 7021 alloys with different contents of Mg and Zn (7021 alloy 1, 2 and 3) was determined (Figure 10). The highest strength and plastic properties were recorded at a welding temperature of 450 °C for all three analyzed alloys. The relative strength of the welds was 86% (alloy 1), 93% (alloy 2) and 85% (alloy 3). Slightly lower values of the relative strength of the welds were obtained for a welding temperature of 500 °C—but only for 7021 alloys 2 (61%) and 3 (66%). In the case of alloy 1, the relative strength of the welds dropped dramatically from 86% to 12%.

The highest UTS in welding trials was achieved for samples from alloy 3 with the highest content of main alloy additions (Figure 10b). However, the increase in the content of alloying elements also translates into a reduction in the plastic flow resistance of the metal during heat extrusion as well as a higher extrusion force and higher mechanical loads acting on the dies. Therefore, in order to ensure an metal flow rate from the die (process efficiency), extrusion force and dimensional tolerances resulting from a small elastic deflection of the die, alloy 2 was adopted for further analysis.

Due to the temperature and speed conditions of the industrial process of the extrusion of hollow sections from 7021 alloy, the temperature in the welding area of 550 °C was assumed for further analysis. For the alloy defined in this way and the welding temperature, the Akeret weldability index was determined, which in this case is *Ϭ_n_/k* = 300 MPa/51 MPa = 5.88. 

Microstructural observations of the samples after the welding process using light microscopy show that, regardless of the process conditions used outside the weld, the grains have a regular, near-axial shape, while in the weld area the grains are elongated (Figure 11, Appendix A in Appendix A). In most cases, no discontinuities were found in the weld area, except for samples of alloys 1 and 2 welded at the lowest temperature of 400 °C, where discontinuities were found along the entire length of the weld (see arrows) (Figure 11a,b). A characteristic feature of the microstructure of alloys 2 and 3 was the occurrence of numerous precipitates within the grains, regardless of the welding process parameters used (Figure 11b,c, Appendix A in Appendix A).

Measurements of the width of the weld zone proved that for alloy 1, the width of the welding area measures approximately 2.5 mm, regardless of the welding temperature. For alloy 2, the width of the welding area measures approximately 1.7 mm for welding temperatures of 400 °C and 450 °C. On the other hand, it is much wider at approximately 2.8 mm after welding at 500 °C. In the case of alloy 3, the welding width was found to increase with increasing temperature; the width of the welding area was 1.77 mm, 2.53 mm and 3 mm for welding temperatures of 400 °C, 450 °C and 500 °C, respectively. 

The grain size measurements proved that, in the alloy 1 samples, the average grain diameter outside the weld was 147–196 µm, with the samples welded at 400 °C having the largest grain size. The average diameter of the grains outside the weld area of the samples welded at 450 °C and 500 °C was comparable and was approximately 150 µm (Figure 12a). The determined average grain size in the welding area is 228–260 µm and, similarly to the outside of the weld, is the largest for samples welded at 400 °C; for the other welding variants, it is comparable (Figure 12a). The coefficient of grain elongation of alloy 1 samples in the weld area is 7.7–9.1 and is the smallest for samples welded at the lowest temperature of 400 °C. In this alloy, fine dispersive precipitations are presented in the microstructure after welding at 450 °C and 500 °C, which inhibited grain growth.

In the alloy 2 samples, the average grain diameter outside the weld is 140–180 µm, with the smallest grain size for samples welded at 500 °C. The determined average grain size in the welding area is 195–266 µm and, similarly to the outside of the weld, is the smallest for samples welded at 500 °C (Figure 12b). The elongation coefficient is 8.5–14.2 and is the highest for samples welded at the lowest temperature of 400 °C. After welding at 450 °C and 500 °C, the grains in the weld area were wider and shorter; characteristically, in the microstructure, the number of dispersive precipitates on the grain cross-section decreased, and the number of obstacles that inhibit the movement of grain boundaries decreased.

Samples from alloy 3 are characterized by the finest grain size, with the average grain diameter outside the weld area being 118–156 µm, while in the welding area, it is comparable for all variants of the welding process and is 124–132 µm (Figure 12c). The elongation coefficient determined is at the level of 6.5–9.2. The microstructure of alloy 3 has the highest number of dispersive precipitates, which inhibit grain growth when exposed to high temperature and pressure during the welding process. The numerous precipitates did not adversely affect the welding process and no discontinuities were found in the welding area of this alloy, regardless of the welding process conditions used (Figure 11, Appendix A in Appendix A).

Figure 13 and Appendix A in Appendix A) show the results of the chemical composition analysis in micro-areas for alloy 2—welding at 450 °C (Figure 13: outside weld and Appendix A in Appendix A: in weld) and 500 °C (Appendix A in Appendix A: outside weld and Appendix A in Appendix A: in weld). Figure 14 shows the Mg content for the alloys tested when welded at 450 °C (Figure 14a) and 500 °C (Figure 14b). Appendix A in Appendix A, in turn, shows the Zn content for all alloys welded at 450 °C and 500 °C (Appendix A in Appendix A). The mean Mg and Zn content for samples welded at 450 °C was comparable in the welding and non-welding areas and averaged 1.75 wt.%. Mg and 6.5 wt.%. Zn (Figure 14a,b). Examination of the chemical composition of the grain cross-section locally indicates the presence of Ti and Zr, with these elements being present in the dispersoids observed inside the grains. Similarly, in samples welded at a higher temperature of 500 °C, the average content of Mg and Zn in the welding area and outside the weld was comparable, being higher than in samples welded at 450 °C and averaging 1.79 wt.% Mg and 6.56 wt.%, Zn (Appendix A in Appendix A).

After welding at 450 °C, the largest variation in Mg content across the grain cross-section was found for alloy 3 (Figure 14). The average Mg content in the non-welded area is 1.48 wt.%, while in the welding area it is 1.59 wt.% and Zn 8.4 wt.% and 9.1 wt.%, respectively. Welding at a higher temperature of 500 °C resulted in an increase in the inhomogeneity of the distribution of the main alloying elements on the cross-sectional area, especially for alloy 3 (Appendix A in Appendix A). The average Mg content in the area outside the weld for this alloy is 1.6 wt.%, while in the welding area it is 1.72 wt.%, Zn, respectively—9.6 wt.%. in the non-welded area and 9.15 wt.% in the welding area. 

Figure 15 and Appendix A in Appendix A show representative images of the fractures of specimens welded at 450 °C and 500 °C, after uniaxial tensile tests. Fractures of samples of all the alloys welded at 450 °C show features of plastic fracture, the proportion depending on the type of alloy. In the case of alloy 1, the characteristic rounded hollows are only locally visible. In the case of alloys 2 and 3, rounded hollows and elevations, characteristic of plastic fracture, are visible on the entire surface. The presence of dimples in the material shows that plastic deformation occurs. Decohesion occurs in the successive parallel slip planes of favorable orientation. As a result of this process, new free surfaces are formed in materials. The shape and size of the dimples are determined by the size and distribution of microstructure discontinuities, such as the micropores, disperse particles and microcracks, plastic properties of the material, and the acting stresses. 

When the specimens were welded at 500 °C, the fracture of the specimens definitely differed from alloy to alloy (Appendix A in Appendix A). In the case of alloy 1, surface decohesion under shear occurred during the uniaxial tensile test, and no signs of plastic deformation are observed. In the case of alloys 2 and 3, holes and elevations characteristic of plastic fracture are visible on the entire surface, with their proportion being greater in alloy 2 (covering the entire surface). The arrows in Figure 15 and Appendix A in Appendix A show the direction of the crack front propagation. The fractures of the specimens after the uniaxial tensile test shown in Figure 15 and Appendix A in Appendix A are consistent with the results of the microstructure observations. Samples showing a predominance of plastic fracture contribution were characterized by good weld quality.

### 3.2. FEM Numerical Calculations 

Figure 16 shows the mean stress distributions in the inlet channels of the porthole die, the welding chambers, and the die opening for die variant 1 and for die variant 2 of the die (left) and the enlarged mean stress distributions in the welding chambers and the die opening of the porthole die (right) during extrusion of the Ø50 × 2 mm tube from EN AW-7021 alloy. Differences between dies can be seen, especially for the inner welding chambers. In general, in the upper regions of the welding chambers (just below the bridge), maximum compressive stress values favorable for material welding are observed in the range 247–267 Mpa depending on the die design, with slightly higher values recorded for die 1. It can be concluded that the maximum values of the average stress in the welding chambers of the porthole dies correspond to the compressive stress value occurring in the laboratory weldability tests (σ_w_ = 300 Mpa). Moving towards the die opening, these values decrease to reach mean stress values close to zero in the die opening itself. In addition, die 2 has lower compressive stresses compared to die 1 that goes from the extrusion axis towards the width of the welding chamber. This level and distribution of the average stress indicate slightly more favorable conditions for the joining of the metal strands in the welding chambers in the case of die 1. In the metal case of the temperature distribution at the height of the welding chamber (Figure 17), comparable temperature values in the range 488–503 °C with a maximum in the die opening are observed for both dies. These values correspond to the welding temperature in the laboratory weldability test (500 °C), for which a moderate relative weld strength of 62% was obtained for 7021 alloy 2. In the case of plastic stress distributions in the welding chambers of porthole die, significantly higher values of this parameter were obtained for die 2: more than 30 for die 2 compared to 20 for die 1 (Figure 18). Additionally, for die 2, there are larger areas with maximum plastic stress values in the volume of the welding chamber, which may indicate more favorable bonding conditions for the metal planes resulting from more intensive metal mixing in the welding chambers. 

Figure 19 presents the distributions described above in numerical notation. Figure 19, bottom right, shows the distribution of the weldability parameter *σ_m_*/*σ_i_* at the height of the welding chambers of die 1 and die 2, i.e., the ratio of the average stress *σ_m_* to the stress intensity *σ_i_*, a parameter that informs about the material welding conditions in the extrusion process through the porthole dies. The higher value of this parameter, the higher probability of producing a high-quality joint in the extruded hollow profile. For comparison, this graph also shows with a dashed line the level of the minimum weldability index from the laboratory weldability tests (*σ_w_*/k = 5.88), which already guaranteed the production of a relatively good quality weld for the analyzed 7021 alloy 2. As can be seen, at the prevailing height of the welding chambers from the sub-bridge space toward the die opening, there are sufficiently favorable welding conditions (determined numerically by FEM) for both dies (*σ_m_*/*σ_i_* of 6–7), which ensure relatively good material susceptibility to extrusion welding—exceeding the minimum weldability index determined by laboratory weldability tests. However, the closer to die opening, the greater the deterioration of the metal welding conditions. 

Based on this distribution, it can be concluded that in the first stage of metal welding, up to approximately half of the height of the welding chamber, more favorable welding conditions occur for die 1, while in the second stage of welding, more favorable welding conditions occur for die 2. In summary, it can be concluded that the influence of die design on the conditions for joining metal strands in the welding chambers is complex. Die 1 provides higher compressive stress values in the welding chambers in absolute terms, while die 2 guarantees better material mixing due to higher plastic stress values in the welding chambers. The weldability parameter indicates more favorable conditions for the metal strands to fuse together for die 1 in the first welding stage and the more favorable conditions for the metal strands to fuse together for die 2 in the second welding stage. Experimental verification under industrial conditions will ultimately indicate which factors that affect the weldability of the metal in the extrusion process are dominant.

### 3.3. Extrusion Trials 

Figure 20 shows photos of Ø50 × 2 mm tubes extruded from 7021 alloy 2 using 2-hole die 1 and 2-hole die 2. The top and middle rows show photos of tubes on the press run out immediately after the metal exits the die opening (Figure 20 top and middle). The bottom row, on the other hand, shows cross-sectional photos of tubes on the cooling table (Figure 20 bottom). From these results, the good surface quality and dimensional accuracy of the extruded products can be inferred. However, detailed studies of the geometric stability of tubes extruded by die 1 (which is the subject of another publication) indicated relatively large dimensional deviations of wall thickness, going beyond the values allowed by the relevant standard [37].

The process parameters recorded during extrusion of the tubes in question indicate a metal discharge velocity from the die hole in the range 3.5–4.5 m/min, depending on the geometry of the porthole die used (Appendix A in Appendix A). Higher values for the metal exit speed were recorded for die 2, which also provided high dimensional accuracy of the extruded tubes. The temperature measured immediately after the metal exited the die opening was in the range of 570–580 °C (Appendix A in Appendix A). Figure 21 shows example macrostructures of the extruded tube using die 1 with 3 longitudinal welds visible in the cross-section. 

Figure 22 shows the microstructures and analyses of the crystallographic orientation images taken from the cross-sectional view of the tube from the weld area and outside the weld area for die 1 and die 2. From observations in the weld area, it can be seen that the shape and grain size do not differ much from those outside the weld area. This is true for both die 1 and die 2 extruded tubes, and the differences are visible in the inverse-pole figure image. In samples taken from tubes extruded on die 1 and die 2, a maximum was found in the <001> direction, indicating an axial texture typical of the extruded products. Minor differences in texture blurring are also evident in the images in the sample taken from the extruded tube on die 1 between the weld and outside the weld. The image of the orientation distribution map in the sample extruded on die 2 differs from the others in local grain growth. At the weld area of this sample, recrystallized grains are visible, probably formed by friction processes during welding. The microstructure of the sample extruded on die 2 also differs in the shape of the grain outside the weld. The grains in this case are elongated.

Grain size distributions were also made from orientation images (Figure 23). Although the grain size is similar in the tested materials and oscillates at approximately 15 µm, the histograms of the grain size distributions are of a different nature. Finer grains were found in the extruded tubes in die 1 at the weld site. In contrast, there are more grains in the 15 to 30 µm range outside the weld. The tube extruded in die 2 showed a particular tendency toward grain growth in the welding. It differed from the others in having a slightly higher average grain size and a greater spread of the grain size range outside the welding area.

Figure 24 shows the mechanical properties of Ø50 × 2 mm tube extruded from 7021 alloy 2 through porthole die 1 (left) and porthole die 2 (right). Mechanical properties were determined in a static tensile test for 3 specimens containing a weld and 1 specimen without a weld. The results indicate high mechanical properties for both dies analyzed—slightly higher tensile properties YS and UTS were obtained for the tube extruded by die 1, while slightly higher plastic properties were obtained for the tube extruded by die 2. Tensile strengths UTS for the unwelded material were obtained at 472 Mpa (die 1) and 476 Mpa (die 2). The average tensile strength Rm for welded specimens is noteworthy at 474 Mpa for the die 1 extrusion variant and at 454 Mpa for the die 2 extrusion variant. This indicates a high susceptibility to welding of 7021 alloy 2 and the production of high-strength longitudinal welds in the extruded tubes—the relative strength of the welds is approximately 100.5% (die 1) and 95.4% (die 2). In the case of the die 2 extrusion variant, highly plasticized material was also obtained in the extruded tubes, both for the samples with a weld (percentage elongation in the range of 17.5–20%) and for the non-welded material (percentage elongation of approximately 17.5%). In the case of the extruded tube using die 1, the percentage of elongation was below 15%.

## 4. Discussion

In this work, an innovative way for predicting weld seam weld quality in round tubes from the 7021 alloy is presented. The method is based on employment of a patented laboratory device for investigation of welding phenomena, allowing full modelling of conditions which are expected to exist in a welding chamber of the porthole die. The 7021 copper-free alloy was chosen because it is relatively easier to extrude with the use of the porthole die—the extrusion force is lower than that for copper-containing alloys, e.g., the 7075 alloy, and the permissible exit speed is higher. The performed investigations include comprehensive tests, beginning from the homogenization of the billets, through laboratory testing of weldability and the industrial test performed on the basis of the preceding FEM simulation of the extrusion process. The three alloys of different contents of Mg and Zn were tested in the welding tests. The increase in the solidus temperature of the investigated alloys as a result of homogenization enables applying a higher heating temperature of the billets that, in turn, decreases the extrusion force and increases the exit speed from the die and, in this way, the productivity of the process increases. The welding tests with the use of the original authors’ device enable determination of the temperature–pressure conditions necessary to obtain good-quality welds. 

The billets from the tested alloys of 7021 grade were first submitted to a homogenization procedure. In all the examined alloys, the low melting components were dissolved during homogenization to a degree sufficient for extrusion practice. As a result, a significant increase in the solidus temperature was achieved, with values within the range from 559.2 °C for alloy 3 to 613.2 °C for alloy 1. The higher solidus temperature enables increasing the billet heating temperature, and so the extrusion force can be decreased. In addition, the exit speed from the die and the resulting effectiveness of the extrusion process can be effectively increased.

In the welding test, three temperatures of 400 °C, 450 °C and 500 °C were applied, whereas the compression pressure was assumed to be stable at the level of 300 MPa. The influence of the welding parameters for alloys with different contents of Mg and Zn (1, 2 and 3) on the mechanical properties of the welds was determined. The highest strength and plastic properties were recorded for a welding temperature of 450 °C for all three analyzed alloys. The relative strength of the welds was 86% (alloy 1), 93% (alloy 2) and 85% (alloy 3). Slightly lower values of the relative strength of the welds were obtained for a welding temperature of 500 °C but only for the alloys 2 (61%) and 3 (66%). In the case of alloy 1, the relative strength of the welds dropped dramatically from 86% to 12%. 

Measurements of the width of the weld zone (Figure 11 and Appendix A and in Appendix A) proved that for alloy 1, the width of the welding area measures approximately 2.5 mm, regardless of the welding temperature. For alloy 2, the width of the welding area measures approximately 1.7 mm for welding temperatures of 400 °C and 450 °C. On the other hand, it is much wider at approximately 2.8 mm after welding at 500 °C. In the case of alloy 3, the welding width was found to increase with increasing temperature; the width of the welding area was 1.77 mm, 2.53 mm and 3 mm for welding temperatures of 400 °C, 450 °C and 500 °C, respectively.

The grain size measurements proved that, in alloy 1 samples, the average grain diameter outside the weld was 147–196 µm, with the samples welded at 400 °C having the largest grain size. The average diameter of the grains outside the weld area of the samples welded at 450 °C and 500 °C was comparable and was approximately 150 µm (Figure 12a). The determined average grain size in the welding area is 228–260 µm and, similarly to the outside of the weld, is the largest for samples welded at 400 °C; for the other welding variants, it is comparable (Figure 12a). In the case of this alloy, fine dispersive precipitations are present in the microstructure after welding at 450 °C and 500 °C, which inhibited grain growth.

In the alloy 2 samples, the average grain diameter outside the weld is 140–180 µm, with the smallest grain size for samples welded at 500 °C. The determined average grain size in the welding area is 195–266 µm and, similarly to the outside of the weld, is the smallest for samples welded at 500 °C (Figure 12b). After welding at 450 °C and 500 °C, the grains in the weld area were wider and shorter; characteristically, in the microstructure, the number of dispersive precipitates on the grain cross-section decreased, and the number of obstacles that inhibit the movement of grain boundaries decreased.

Samples from alloy 3 are characterized by the finest grain size, with the average grain diameter outside the weld area being 118–156 µm, while in the welding area, it is comparable for all variants of the welding process and is 124–132 µm (Figure 12c). The microstructure of alloy 3 has the highest number of dispersive precipitates, which inhibit grain growth when exposed to high temperature and pressure during the welding process. 

The fracture of the welded samples was also submitted to microscopic observations (Figure 15 and Appendix A in Appendix A). The figures show representative images of the fractures of specimens welded at 450 °C and 500 °C, after uniaxial tensile tests. Fractures of samples of all the alloys welded at 450 °C show features of plastic fracture, the proportion depending on the type of alloy. In the case of alloy 1, the characteristic rounded hollows are only locally visible. In the case of alloys 2 and 3, rounded hollows and elevations, characteristic of plastic fracture, are visible on the entire surface. The presence of dimples in the material shows that plastic deformation occurs. When the specimens were welded at 500 °C, the fracture of the specimens definitely differed from alloy to alloy. In the case of alloy 1, surface decohesion under shear occurred during the uniaxial tensile test, and no signs of plastic deformation are observed. In the case of alloys 2 and 3, holes and elevations characteristic of plastic fracture are visible on the entire surface, with their proportion being greater in alloy 2 (covering the entire surface). The fractures of the specimens after the uniaxial tensile test are consistent with the results of the microstructure observations. Samples showing a predominance of plastic fracture contribution were characterized by good weld quality.

The FEM simulations were performed for two porthole dies, which differ in the geometry of the inlet channels and welding chambers (Figure 16, Figure 17 and Figure 18). The calculations indicated differences in the distributions of predicted welding parameters, such as mean stress, temperature and plastic strain. In general, in the upper regions of the welding chambers (just below bridge), the maximum compressive stress values are observed in the range 247–267 MPa depending on the die design. The slightly higher values were obtained for die 1. It can be seen that the maximum values of the mean stress in the welding chambers of the porthole dies correspond to the compressive stress value occurring in the laboratory weldability tests (*σ_w_* = 300 MPa). In the case of temperature distribution at the height of the welding chamber (Figure 19), comparable values within the range of 488–503 °C with a maximum in the die opening are observed for both the dies tested. These values are slightly higher than that where the maximal relative strength was obtained in the laboratory test and correspond to a welding temperature of 500 °C, for which a moderate relative weld strength of 62% was obtained for 7021 alloy 2. In the case of plastic strain distributions in the welding chambers of the porthole die, significantly higher values were obtained for die 2: above 30 for die 2 compared to 20 for die 1. Additionally, in the case of die 2, there is a larger area of maximum plastic strain values in the volume of the welding chamber, which may indicate more favorable bonding conditions resulting from the more intensive mixing of the metal in the welding chambers.

The distribution of the welding parameters *σ_m_*/*σ_i_* in the welding chambers is also important. In the laboratory weldability tests, an *σ_w_*/k value equal to 5.88 guaranteed the production of a relatively good quality weld for alloy 2. As can be seen, at the prevailing height of the welding chambers from the sub-bridge space toward the die opening, there are sufficiently favorable welding conditions (determined numerically by FEM) for both dies (*σ_m_*/*σ_i_* of 6–7), which ensures relatively good material susceptibility to extrusion welding. The influence of die design on the conditions for joining metal streams in the welding chambers is inconclusive. Die 1 provides higher compressive stress values in the welding chambers, whereas die 2 guarantees better material mixing due to higher plastic strain values in the welding chambers. The weldability parameter indicates more favorable conditions for the metal streams to join together for die 1 in the first welding stage and the more favorable conditions for joining together for die 2 in the second welding stage (Figure 19).

Industrial extrusion tests were performed using alloy 2. Round tubes of Ø50 × 2 mm were extruded through porthole dies 1 and 2 as described in the FEM simulations. The simulation provided useful information for proper selection of extrusion parameters and die geometry. The exit temperature and exit speed were recorded during the experiment. The higher permissible exit speed was obtained for die 2 at the level of 4.5 m/min compared to 3 m/min for die 1. The temperature measured immediately after the metal exited the die opening was in the range of 570–580 °C. The measurements of the dimensional accuracy of the extruded tubes indicated better behavior of die 2.

The FEM simulation and the industrial trials indicated the important role of the porthole die geometry when extruding the 7000 series alloys. Die 1 was based on the variant usually used during extrusion of the 6000 series alloys, whereas die No 2 was the modified version elaborated in FEM simulation of the process. Finally, it can be noticed that the die design for the 7000 series alloys and particularly for the 7021 alloy differs considerably from that of die 1. Despite the similarity in the inlet channel layout, the channels should be wider while the bridges must be higher and thicker to withstand the increased extrusion force and to avoid elastic deflection of the die. The height of the welding chamber should be higher. The die insert should be equipped with pockets, which regulate metal flow in the region close to the die opening. In such a case, the metal flow is more uniform and this is what improves the geometrical stability and dimensional accuracy of the extruded profile.

EBSD studies of the microstructure of the industrially extruded tubes proved the variation in grain size and shape at the weld site and outside the weld for both die 1 and die 2 extruded tubes, with die 1 extruded tubes having a more homogeneous microstructure in the cross-section (Figure 22). The average grain size of the tubes tested is approximately 15 µm. For tubes extruded by die 1, grains of less than 20 µm predominate, with grains in the 20–40 µm range more frequently observed in the area outside the weld (Figure 23). In the case of tube extruded by die 2, when using a higher extrusion speed (4.5 m/min), the grains have a varied shape in the area outside the weld. In the microstructure, there are both evenly spaced and elongated grains with clearly bulged boundaries, indicating the occurrence of structure renewal processes. In the weld area, the grains are evenly spaced, with locally large recrystallized grains, which were probably formed during the deformation process due to significant heating of the material (Figure 22). In the microstructure of the tube extruded by die 2, grains up to 80 µm are present, with grains of less than 30 µm predominating. In addition, the histogram showing the grain distribution in the area outside the weld is more flattened than that showing the weld area (Figure 23).

The mechanical properties of the extruded tubes were determined in the tensile test. The results indicate high mechanical properties for both the dies analyzed–higher strength properties R_0.2_ and R_m_ were obtained for the tube extruded with die 1, while slightly higher plastic properties were obtained for the tube extruded with die 2. The R_m_ for the unwelded material was obtained at 472 MPa (die 1) and 476 MPa (die 2). The average R_m_ for welded specimens is as high as 474 MPa for die 1 and 454 MPa for die 2. This indicates a very high susceptibility to welding of 7021 alloy 2 that guaranties production of high-strength longitudinal welds in the extruded tubes—the relative strength of the welds is approximately 100.5% (die 1) and 95.4% (die 2). The relative strength of the tubes is higher in comparison to that obtained in the laboratory test but it can be explained by more intensive plastic deformation in the welding chambers during the extrusion process. The slightly lower Rm values and higher A values for tubes extruded by die 2 can be explained by the local grain growth in the weld area, as well as the larger grain size compared to tubes extruded by die 1 (Figure 24).

Comparison of the strength properties of samples with a longitudinal weld produced in the extrusion process of the AlZnMg alloy 2 and the numerically predicted FEM level of compressive stress in the die welding chamber, determined by the die construction, shows a clear relationship. An approximately 8–10% higher level of compressive stress in the welding chamber for die 1 translates into an approximately 5% higher UTS for the product extruded on this die. Moreover, FEM predicted higher values of the plastic strain in the welding chamber of die 2 resulting in a higher percentage elongation for samples with the weld. This allows us to conclude that the FEM results and the results of extrusion experiments are in high agreement, at least on a qualitative level. A high quantitative consistency can be found between the weldability index determined for alloy 2 in weldability tests (*б_w_*/k = 5.88) and numerically determined by FEM for both dies (*б_w_*/k of 6–7). Both dies ensured obtaining this indicator, which was confirmed by the results of the static tensile test (high-quality seam welds of high strength). Higher values of the weldability index for matrix 1 were obtained in the first stage of metal welding in the welding chamber, which proves that this initial stage of welding determines the quality/mechanical strength of the welds. 

## 5. Conclusions

It is possible to produce the high-quality weld seams in the hollow profiles extruded from the AlZnMg alloys through the porthole dies. The UTS in the T5 temper for extruded tubes of Ø50 × 2 mm from the 7021 alloy 2 with weld seams in relation to the die used was as high as 454–474 MPa, whereas the relative strength of the welds was within the range of 95 ÷ 100.5 MPa. Worth noting is the high plasticity of samples with the welds—the percentage elongation for die 2 was within the range of 17.5–20%. The optimum welding temperature that produces the best mechanical strength is obtained at 450 °C. This is possible due to proper selection of the chemical composition of the 7021 alloy (Zn content at the level of 5.47% and Mg of 2.12%) as well as appropriate selection of stress–temperature conditions in the extrusion process, e.g., a unit pressure at the level of 300 MPa resulting from the die design and a temperature of 550 °C determined by process conditions.Predicted through FEM, higher values of compression stress in the welding chamber of the porthole die 1 resulted in higher UTS for samples with the weld from 7021 alloy 2. In turn, FEM predicted higher values of the plastic strain in the welding chamber of die 2, resulting in a higher percentage elongation for samples with the weld.The proposed method for predicting the quality of welds in the extruded hollow profiles from the aluminum alloys is based on the laboratory tests of weldability with the use of patented device, which simulates bonding conditions in the welding chamber of the porthole dies. At first, the weldability factor б_n_/k of Akeret is determined, and next, it is verified numerically for the extrusion process. The performed analysis confirmed the possibility of obtaining a factor value of б_n_/k = 5.88 for both the dies examined. This result was verified positively in the industrial trials of the extrusion of tubes of Ø50 × 2 mm from alloy 2. The obtained industrial trial values of the relative strength of welds are higher compared to those predicted in the laboratory tests and can be explained by the positive influence of the intensive plastic mixing of the material within the inlet channels and in welding chambers.Examination of the microstructure of specimens welded in a device for weldability tests has shown that, regardless of the parameters of the welding process and the chemical composition of the alloy, the grains in the area of welding have an elongated shape (heavy sheared regions where the samples undergo welding), while grains have a regular, near-axial shape outside the area of welding. The width of the weld area depends on the alloy type and welding process conditions and is approximately 2.5 mm for alloy 1, 1.7–2.8 mm for alloy 2 and 1.77–3 mm for alloy 3. In most of the samples tested, no discontinuities in the welding area were found. Investigations of the chemical composition in the welding area and outside the welding area also showed no differences in the content of the main alloying elements Mg and Zn.Studies of the microstructure of tubes made of alloy 2 extruded in industrial conditions showed greater microstructure differentiation in the weld area and outside the weld area in the case of extrusion through porthole die 2. For tubes extruded by die 2, in the weld area, the grains are similar in shape to parallel, with locally large, recrystallized grains. The variation in microstructure on the cross-section of the tube extruded by die 2 influenced the mechanical properties. The UTS for the unwelded material was as high as 472 MPa for die 1 and 476 MPa for die 2, whereas, for welded specimens, is as high as 474 MPa for die 1 and 454 MPa for die 2. The slightly lower UTS values and higher elongation values for tubes extruded by die 2 can be explained by the local grain growth in the weld area, as well as by the slightly larger grain size compared to tubes extruded by die 1.

Future research will focus on the possibilities of further maximizing the metal exit speed during extrusion of AlZnMg alloys, as these alloys are generally difficult to deform. Solutions will be sought that will ensure a consensus between the quality of longitudinal welds in extruded products (mechanical strength), its dimensional tolerances and the efficiency of the extrusion process.

## Figures and Tables

**Figure 1 materials-16-05817-f001:**
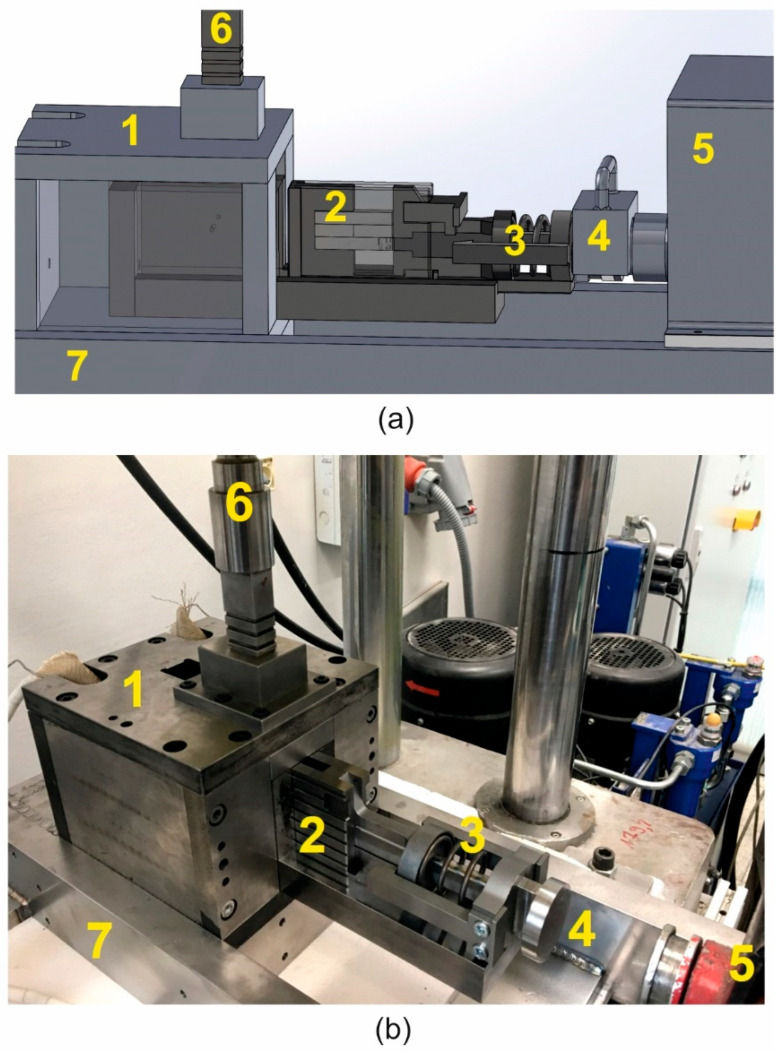
Device for weldability tests of metals and alloys: 3D model visualization (**a**) and research laboratory device (**b**); 1—heating chamber, 2—shearing and welding cartridge tool, 3—compression hydraulically driven stem with presser, 4—adapter of hydraulic cylinder and tool assembly holder, 5—hydraulic cylinder, 6—upper shearing punch, and 7—main construction ram (Patent No PL230273B1, AGH Krakow, 2018 [36]).

**Figure 2 materials-16-05817-f002:**
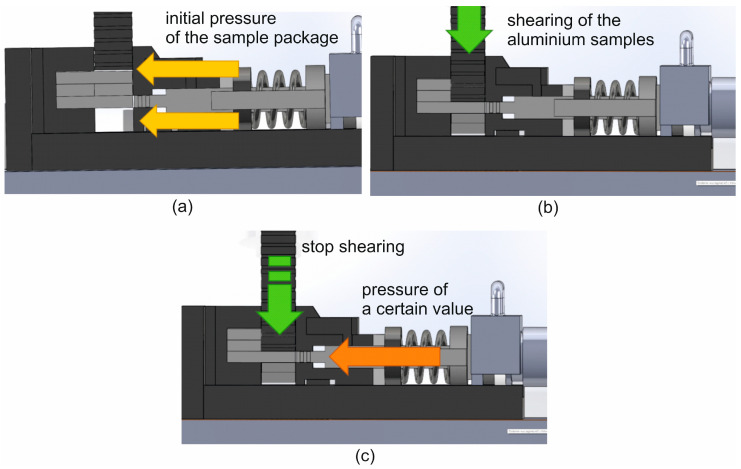
Individual stages of the weldability test: (**a**)—the tool cassette is inserted into the heating chamber and the presser presses the sample package, (**b**)—the samples are sheared with the upper punch while holding the presser continuously, and (**c**)—the shearing punch stops and the pressing punch applies a certain pressure to the welded samples.

**Figure 3 materials-16-05817-f003:**
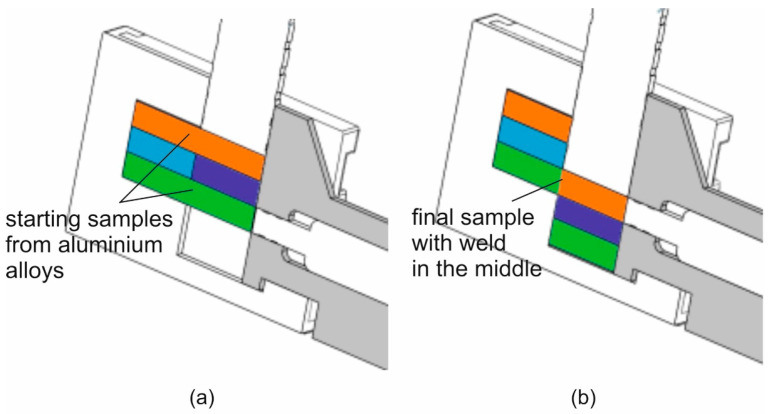
Arrangement of samples and counter-samples in the toolbox in the starting position (**a**) and in the final position after the end of the tests (**b**); orange and green colours mean samples from aluminium alloy; blue and purple colours mean counter-samples from steel.

**Figure 4 materials-16-05817-f004:**
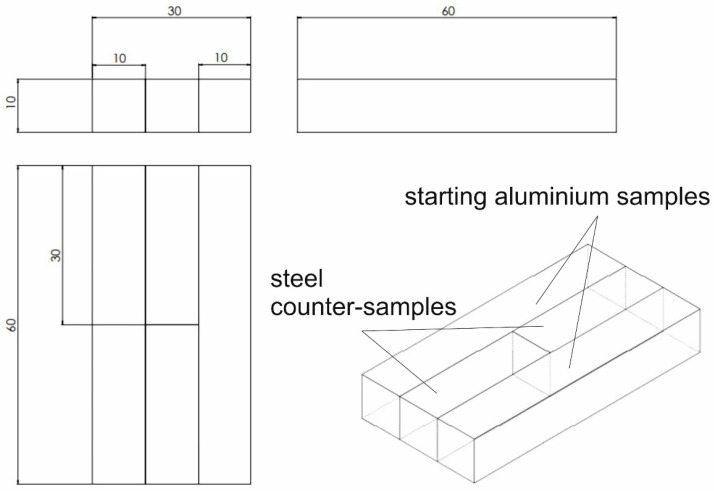
Dimensions of samples and counter-samples in the starting position.

**Figure 5 materials-16-05817-f005:**
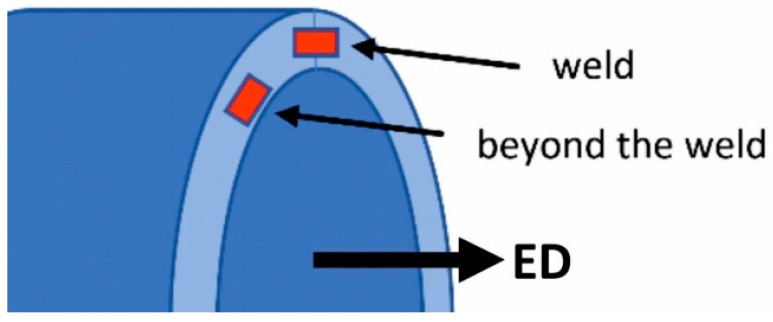
The method of taking samples from the extruded tube for further examination of the structure and mechanical properties.

**Figure 6 materials-16-05817-f006:**
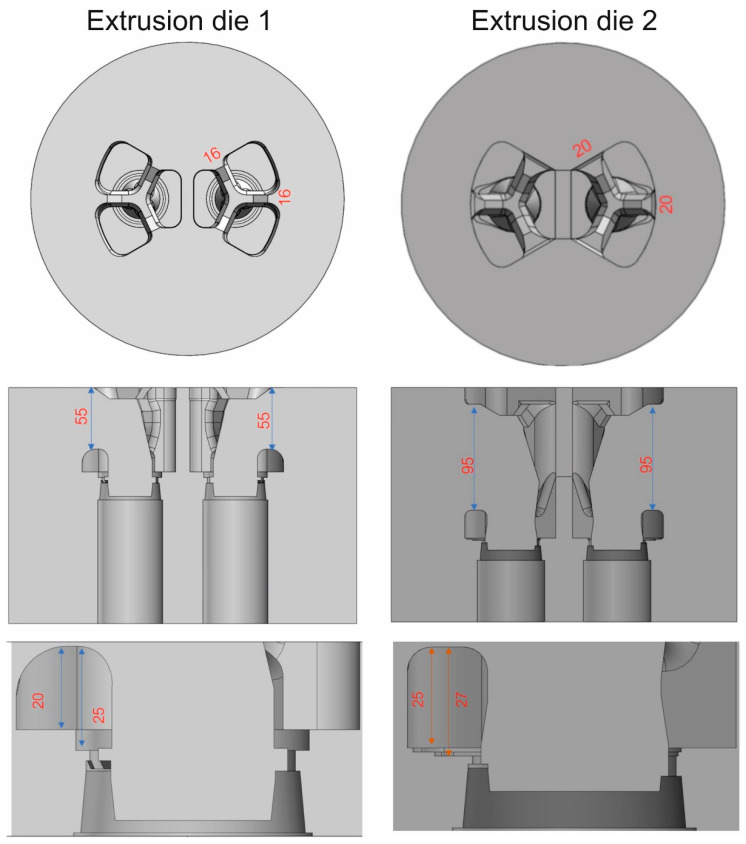
Dimensions of different porthole dies for extrusion of tubes of Ø50 × 2 mm from 7021 alloy 2—top view and the cross-sectional view.

**Figure 7 materials-16-05817-f007:**
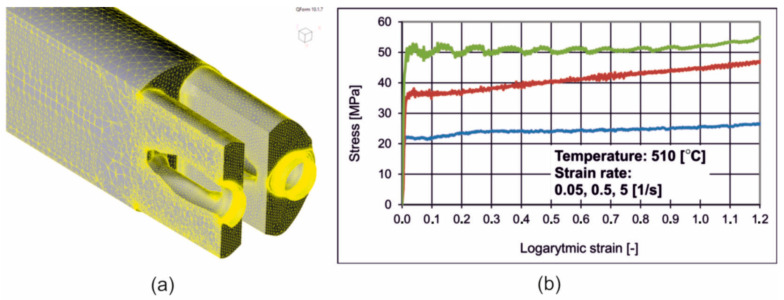
The geometry of the extruded material (**a**) and FEM material model indicating the dependence of plastic stress on logarithmic strain for different strain rates for alloy 2 (**b**); blue colour for 0.05 1/s, red colour for 0.5 1/s and green colour for 5 1/s [37].

**Figure 8 materials-16-05817-f008:**
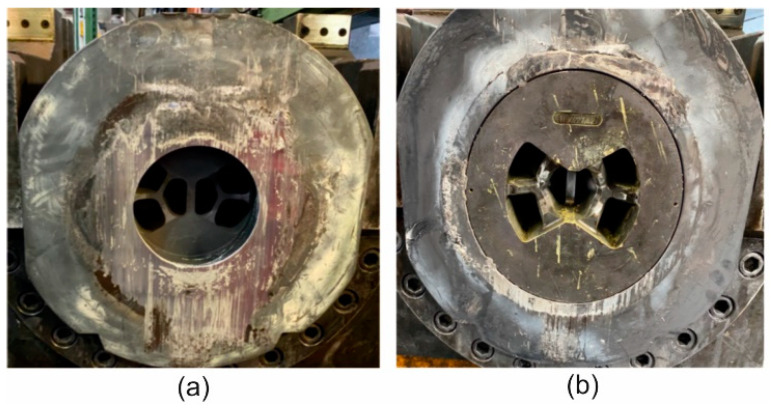
Porthole extrusion dies used in experimental trials of the extrusion of tubes of Ø50 × 2 mm from aluminum 7021 alloy 2 by using the 7-inch 2500 T hydraulic press: (**a**)—die 1; (**b**)—die 2.

**Figure 9 materials-16-05817-f009:**
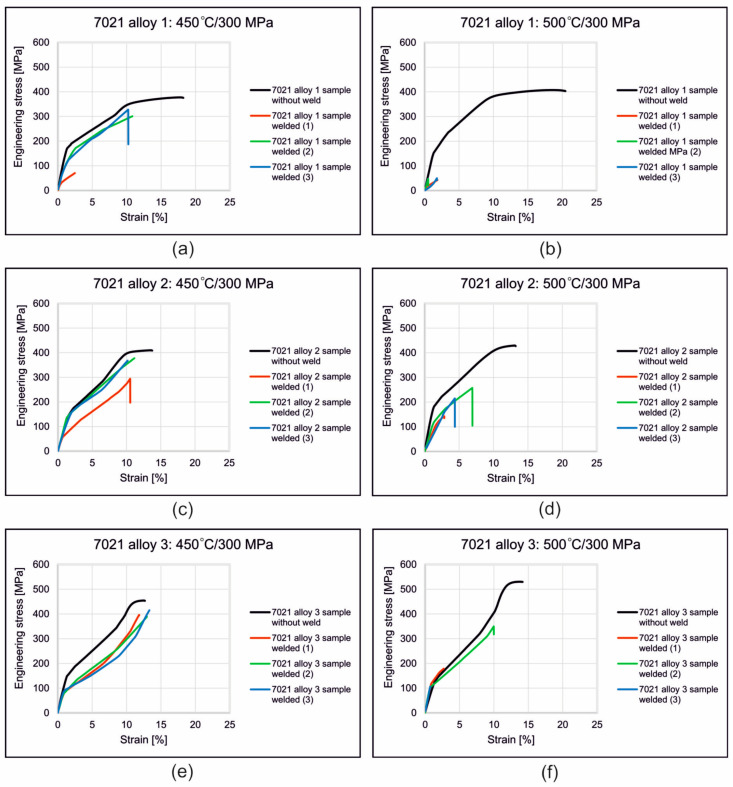
Stress–strain curves recorded during static tensile testing of samples welded from alloys 1, 2 and 3 at 450 °C and 500 °C; 7021 alloy 1: welding temperature 450 °C/welding pressure 300 MPa (**a**) 7021 alloy 1: welding temperature 500 °C/welding pressure 300 MPa (**b**) 7021 alloy 2: welding temperature 450 °C/welding pressure 300 MPa (**c**) 7021 alloy 2: welding temperature 500 °C/welding pressure 300 MPa (**d**) 7021 alloy 3: welding temperature 450 °C/welding pressure 300 MPa (**e**) 7021 alloy 3: welding temperature 500 °C/welding pressure 300 MPa (**f**).

**Figure 10 materials-16-05817-f010:**
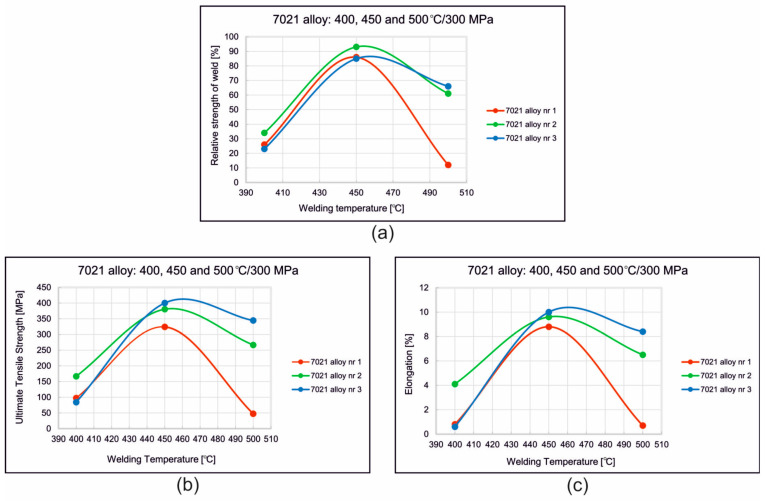
Influence of welding temperature on the relative strength of welds (**a**), the ultimate tensile strength (UTS) of samples (**b**) and elongation (**c**) for 7021 alloys with different chemical compositions.

**Figure 11 materials-16-05817-f011:**
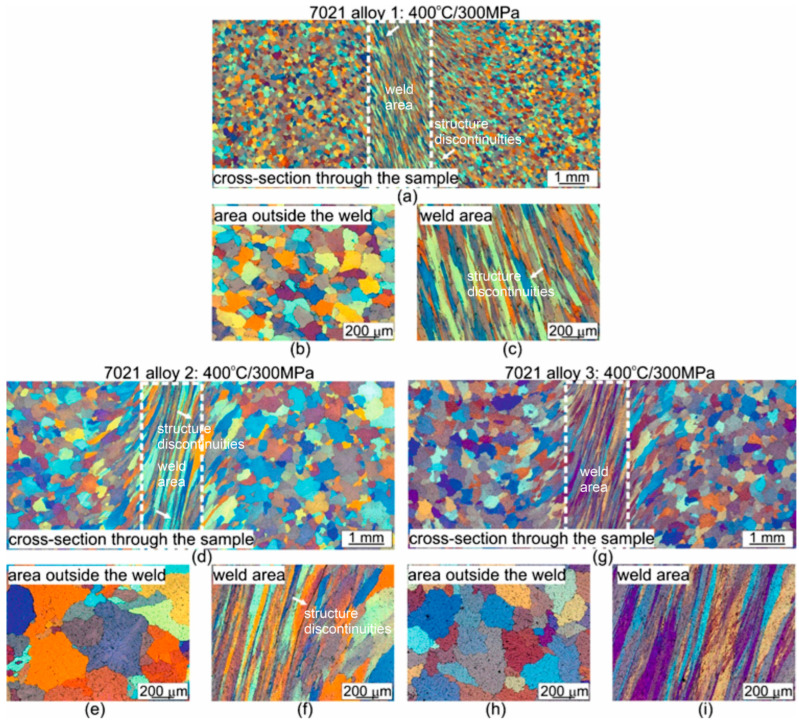
Microstructure of welded 7021 alloy in relation to chemical composition; welding was performed under identical process parameters: *T* = 400 °C, *p* = 300 MPa; (**a**–**c**) 7021 alloy 1: 1.20% Mg, 5.27% Zn; (**d**–**f**) 7021 alloy 2: 2.12% Mg, 5.47% Zn; (**g**–**i**) 7021 alloy 3: 2.12% Mg, 8.02% Zn; light microscopy.

**Figure 12 materials-16-05817-f012:**
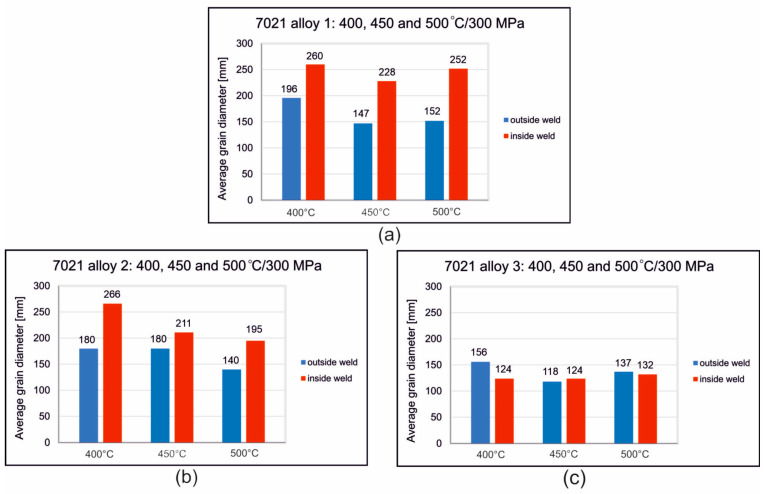
Average grain size outside and inside the weld; (**a**) 7021 alloy 1: 1.20% Mg, 5.27% Zn; (**b**) 7021 alloy 2: 2.12% Mg, 5.47% Zn; (**c**) 7021 alloy 3: 2.12% Mg, 8.02% Zn.

**Figure 13 materials-16-05817-f013:**
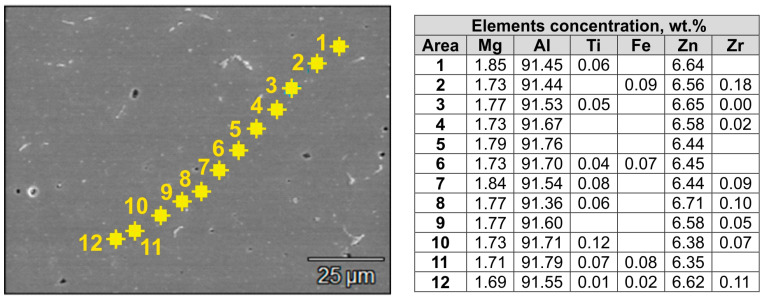
Microstructure of alloy 2 in the area outside the weld and results of the chemical composition test on the grain cross-section; welding process conditions: *T* = 450 °C and *p* = 300 MPa; SEM/EDS.

**Figure 14 materials-16-05817-f014:**
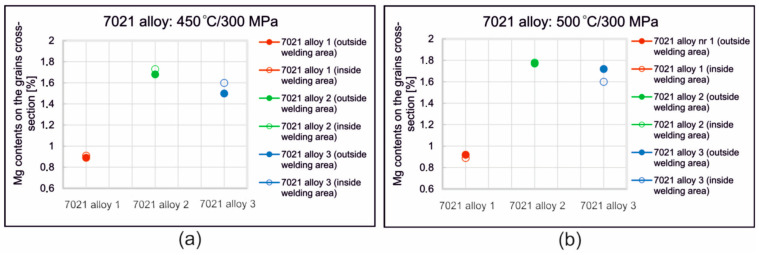
Average Mg content on the grain cross-section in the welding area and outside the welding area for the 7021 alloy 2 tested: (**a**) *T* = 450 °C, *p* = 300 MPa; (**b**) *T* = 500 °C, *p* = 300 MPa.

**Figure 15 materials-16-05817-f015:**
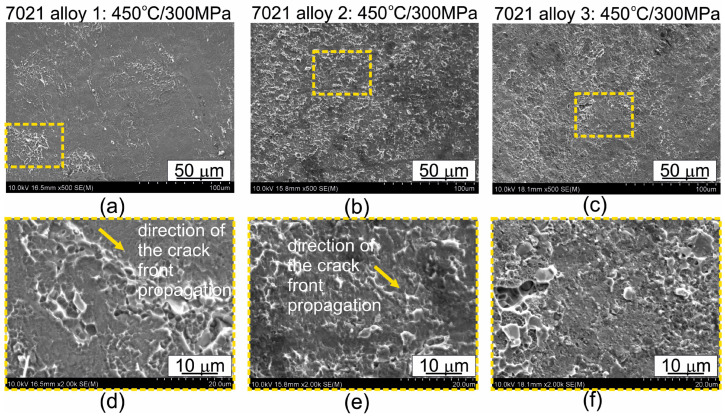
Fracture surfaces after uniaxial the tensile test; welding was performed under identical process parameters: *T* = 450 °C, *p* = 300 MPa; (**a**,**d**) 7021 alloy 1: 1.20% Mg, 5.27% Zn; (**b**,**e**) 7021 alloy 2: 2.12% Mg, 5.47% Zn; (**c**,**f**) 7021 alloy 3: 2.12% Mg, 8.02% Zn. Figures (**d**–**f**) are the enlarged areas indicated by the dashed rectangles in Figures (**a**–**c**).

**Figure 16 materials-16-05817-f016:**
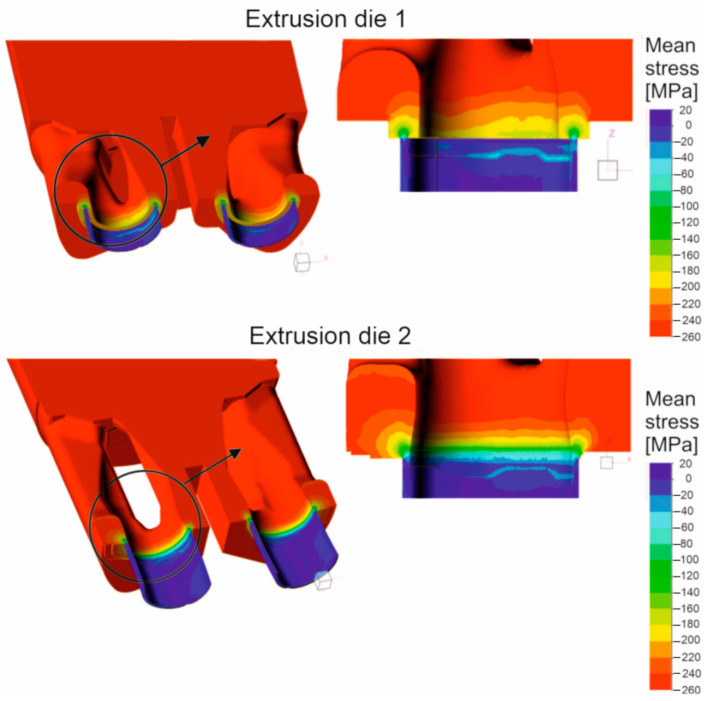
Distribution of mean stress during extrusion of tubes of Ø50 × 2 mm from 7021 aluminum alloy through porthole dies of different geometries—extrusion die 1 and extrusion die 2 (results of FEM calculations).

**Figure 17 materials-16-05817-f017:**
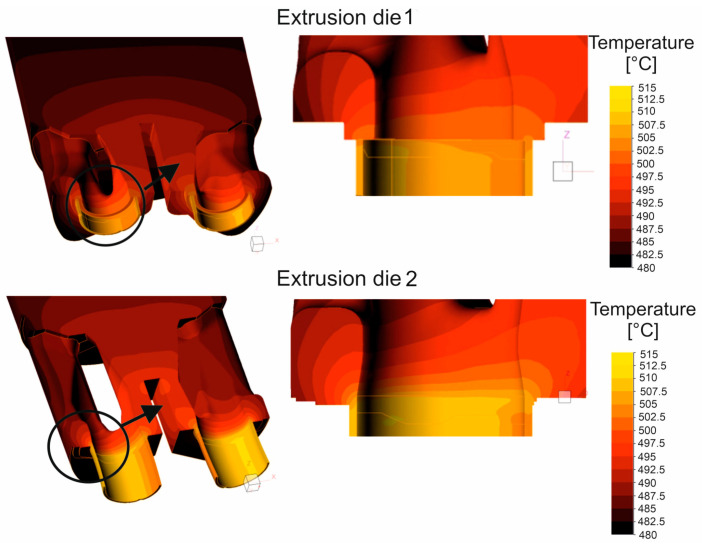
Distribution of temperature during extrusion of tubes of Ø50 × 2 mm from 7021 aluminum alloy through porthole dies of different geometries—extrusion die 1 and extrusion die 2 (results of FEM calculations).

**Figure 18 materials-16-05817-f018:**
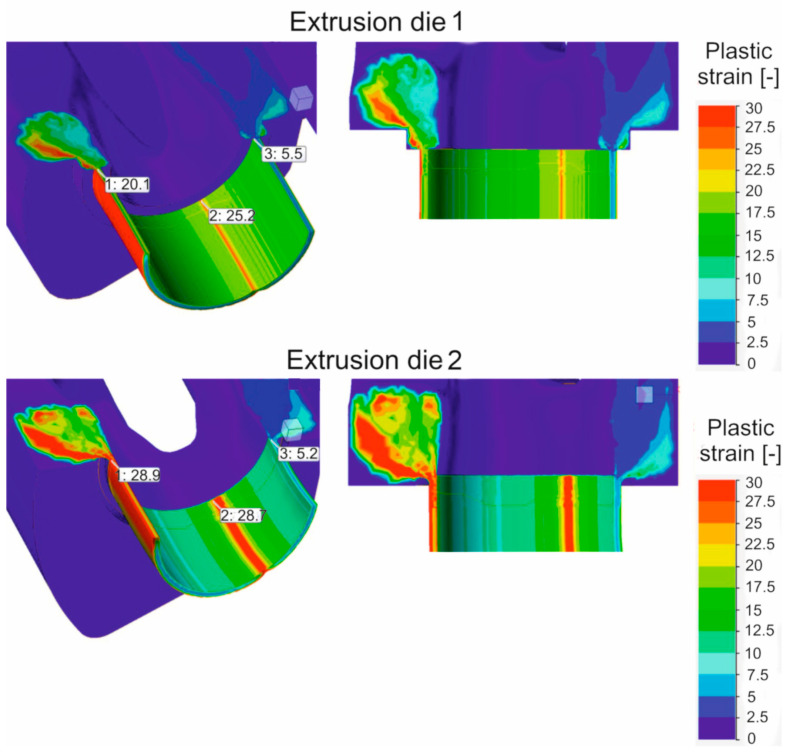
Distribution of plastic strain during extrusion of tubes of Ø50 × 2 mm from 7021 aluminum alloy through porthole dies of different geometries—extrusion die 1 and extrusion die 2 (results of FEM calculations).

**Figure 19 materials-16-05817-f019:**
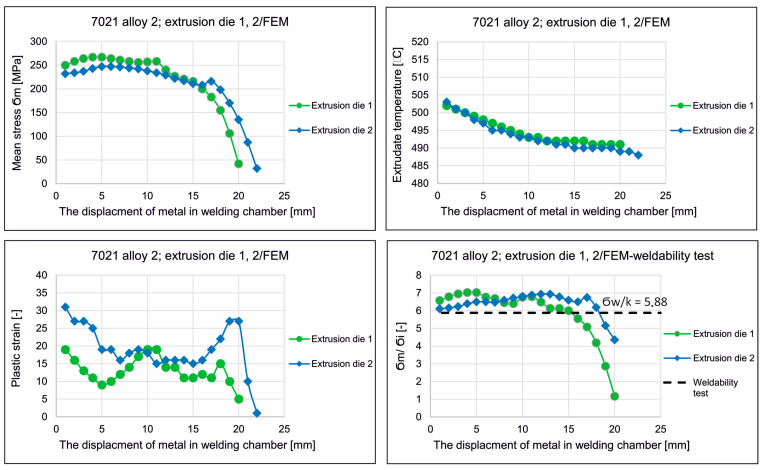
FEM numerically predicted the relationship between the displacement of metal in the welding chamber and the mean stress *σ_m_*, extrudate temperature, plastic strain and weldability index *σ_m_*/*σ_i_* during extrusion of Ø50 × 2 mm tubes from 7021 aluminum alloy 2.

**Figure 20 materials-16-05817-f020:**
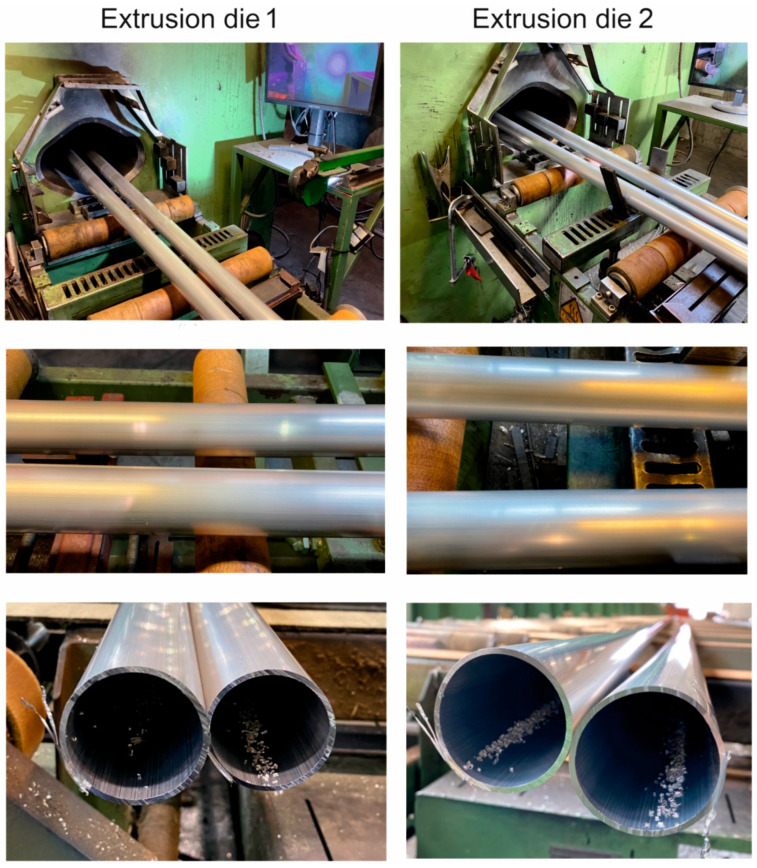
Extruded tubes of Ø50 × 2 mm from 7021 alloy 2 on the press run-out for dies of different geometries: die 1; *T*_0_ = 480 °C, *V*_1_ = 3.5 m/min, die 2; *T*_0_ = 480 °C, *V*_1_ = 4.5 m/min.

**Figure 21 materials-16-05817-f021:**
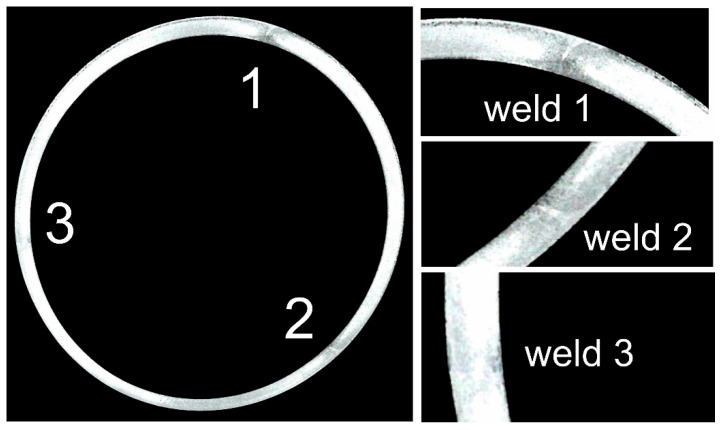
Example macrostructures of a tube of Ø50 × 2 mm extruded through die 1 from 7021 alloy 2 revealing the locations of welds in the cross-section.

**Figure 22 materials-16-05817-f022:**
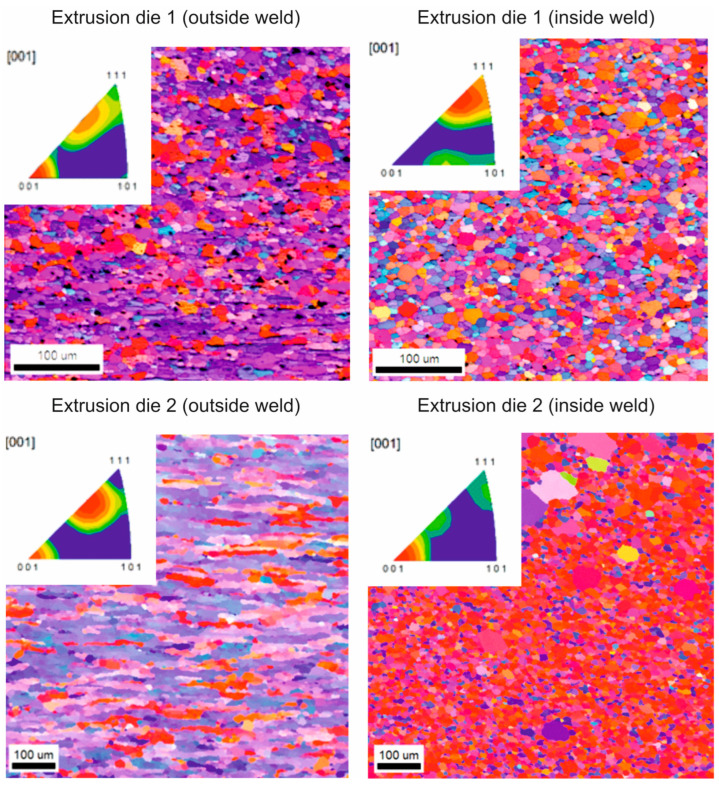
Microstructures and crystallographic orientation images taken from the cross-sectional view of the tube of Ø50 × 2 mm extruded from 7021 alloy 2 from the weld area and outside the weld area for die 1 and die 2.

**Figure 23 materials-16-05817-f023:**
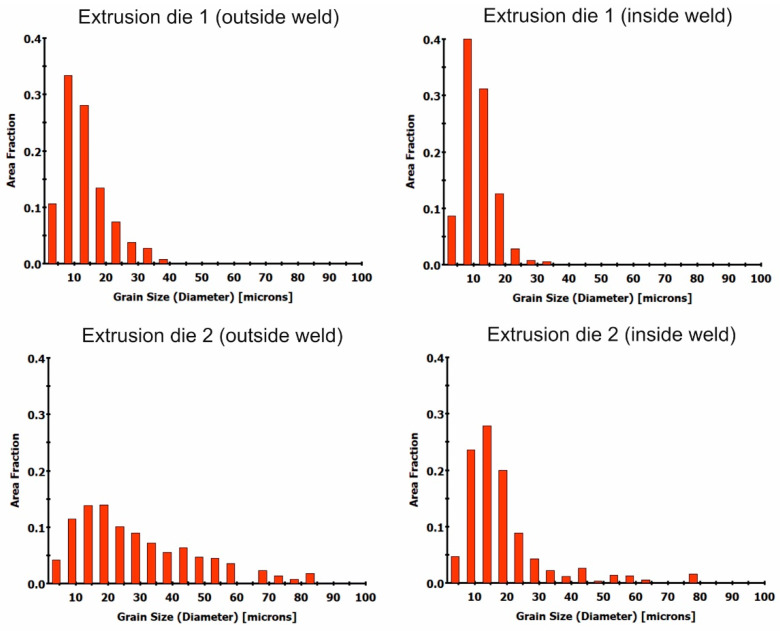
Grain size distributions for material outside the weld and inside the weld for tubes extruded by using die 1 and die 2 (tube of Ø50 × 2 mm extruded from 7021 alloy 2).

**Figure 24 materials-16-05817-f024:**
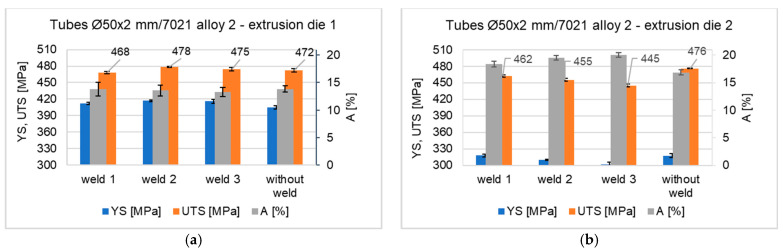
Mechanical properties of Ø50 × 2 mm tube extruded from 7021 alloy 2 through porthole die 1 (**a**) and porthole die 2 (**b**) determined in the static tensile test.

**Table 1 materials-16-05817-t001:** The chemical composition of investigated EN AW-7021 alloys, mass percentage.

Alloy Denotation	Si	Fe	Cu	Mg	Cr	Zn	Ti	Zr
7021 alloy 1	0.09	0.22	0.00	1.20	0.00	5.27	0.01	0.15
7021 alloy 2	0.08	0.21	0.00	2.12	0.00	5.47	0.01	0.15
7021 alloy 3	0.09	0.22	0.00	2.06	0.00	8.02	0.02	0.15

**Table 2 materials-16-05817-t002:** DSC test results of as-cast and homogenized EN AW-7021 alloys.

Alloy	Solidus Temperature, °C	Incipient Melting Heat, J/g
7021 alloy 1	611.8	-
7021 alloy 2	478.1	0.07
7021 alloy 3	478.2	0.68
7021 alloy 1 (homogenized)	613.2	-
7021 alloy 2 (homogenized)	572.1	0.29
7021 alloy 3 (homogenized)	559.2	-

**Table 3 materials-16-05817-t003:** The coefficients from the Hensel–Spittel equation.

A	m_1_	m_2_	m_3_	m_4_	m_5_	m_7_	m_8_	m_9_
1090	−0.0675	−0.055	0.21	−0.015	0.019	−0.026	−0.0019	0.056

**Table 4 materials-16-05817-t004:** The defined parameters of the FEM modelled extrusion process of tubes of Ø50 × 2 mm from 7021 alloy 2.

Alloy	7021 alloy 2
Billet dimensions	Ø178 × 700 mm
Billet temperature	480 °C
Container/Die temperature	465 °C
Extrusion ratio	42
Stem velocity	1.58 mm/s
Metal exit speed	3–4.5 m/min
Friction coefficient	m = 1

## Data Availability

Not applicable.

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
