# Peer review of "FEM Numerical and Experimental Work on Extrusion Welding of 7021 Aluminum Alloy"

_materials, 2023, doi:10.3390/ma16175817_

Round 1

Reviewer 1 Report

TITLE OF PAPER:

Research on Extrusion Welding of 7021 Aluminium Alloy

REVIEWER’S COMMENTS :

The paper deals with extrusion welding of 7021 aluminium alloy at different welding temperatures and constant pressure of 400 MPa. This paper is of an acceptable quality and can be published. However, some minor corrections are required to improve the paper as follows:

1.   Microstructures of the weld region in Fig. 10(c),(f),(i) show large, elongated and bent grains with the direction of grain elongation is nearly perpendicular to the axial pressure direction. However, the weld metal microstructure in other pressure welding processes which have the same weld formation mechanism such as RFW is usually marked by the presence of fine grained equiaxed structure in the weld zone due to recrystallization whereas the elongated grains are categorized as TMAZ. Please, add some more explanation related to these different results.

2.   The optimum welding temperature which produces the best strength is obtained at 450 oC. This finding should be added to Conclusions.

3.   Please add error bars to the bar charts in Fig. 13 and Fig. 31.

4.   Despite it is not compulsory, the additional work on microhardness distribution across the weld joints is important to gain better understanding to the mechanical properties of the weld joints.

Author Response

Thank you for your comments.

Reviewer 2 Report

This is a timely effort by authors on "Research on Extrusion Welding of 7021 Aluminium Alloy". However, there are few suggestions to improve this manuscript as this manuscript has many jargons. 

1. Title of this manuscript must be improved to reflect both numerical and empirical work on extrusion welding. 

2. Novelty needs to be highlighted in a better way.

3. Introduction section contains mostly (93.02% of total references) outdated references. All the references must be latest I.e. published within last 5years as currently no. Of outdated references are 40. 

4. Table 2 & 3 should be merged into one table.

5. Materials and Methods must be improved with better nomenclature regarding extrusion welding process. Currently, weld testing procedure was explained with better autonomy.

6. Also what is steel counter samples doing there in Al alloy weld testing?

7. The details of FEM modelling are incomplete too as there no information about element type, element size, mesh sizing, mesh verification, etc.

8. The dimensions of weld samples should be shown clearly. And did you use the standardized sample dimensions. 

9. Results section needs much improvements in terms of following comments. 

10. The highest strength should be achieved for alloy 3, as can be seen from Fig. 8e and 9b?

11.  Figure 10 must be captioned with arrows pointing within the figures instead of lengthy captions after figure numbers. And same should be implemented for Figs.  11 and 12.

12. How did you measure the Mg and Zn content from chemical composition test? Plz write it in the materials and methods.

13. Figs. 11 & 12 and 15-17, 19, 21 & 27 must go in supplementary file. Please do the needful. 

14. So, how do you conclude the degree of agreement between numerical and empirical studies?

15. Please mention clearly the dimensions of dies 1 and 2 as per good dimensioning standard. 

16. Discussion section is short and must be enlarged to accommodate all the findings with their reasons. 

17. Also what is the rational behind selecting 300 MPa as constant pressure?

18. Are all the testing procedures standardized or literature based?

19. Plz write the future work after conclusion section.

Author Response

Thank you for your comments.

Author Response

Thank you for your comments.

Round 2

Reviewer 2 Report

The authors have now improved the manuscript, however,  this manuscript needs more improvements which insists me to request authors for bothering further and following changes;

1. All the explanations to my previous suggestions must go into the main manuscript. For instance, please ensure that you have included the FEM relevant explanations regarding node and finite element numbers, mesh sizes, etc. In the main manuscript at appropriate place. Please do the needful for all of my previous comments. 

2. Please use solidworks for dies and samples dimensions as per the ASME standard of dimensioning. 

3. The references have been updated but they are reduced to less than 30. The recent and relevant references should be increased up to 35.

Minor editing of grammatical and spelling mistakes are observed especially in the report of authors to reviewers. This should be corrected as this will become the part of main manuscript hereafter. 

Reviewer 3 Report

Dear authors,     I read an interesting paper on "FEM numerical and experimental work on extrusion welding of 7021 aluminum alloy" but rejected it for the following reasons.     For example, the     [1] p. 1 L. 42 "AlMg3-5": does this notation represent the composition of Mg? Please describe it in a way that is easy for the reader to understand. [2] p. 3 L. 103-104: It seems strange that the topic of Mg-Al-Zn alloys suddenly appears in the text when the paper is about AlZnMg alloys. I feel it is necessary to explain why the example of Mg-Al-Zn alloys is mentioned. [3] p. 3 L. 142 "Akeret indicator σn/k": What does the Akeret indicator mean? Please cite references or explain in the text. [4] p. 5 L. 180-189: I think the numbers 1, 2, and 3 in the text should be aligned with the symbols (a), (b), and (c) in the figure in Figure 2. [5] p. 6 Section 3.2: The text should also describe the dimensions of the tensile specimen, the strain rate during tensile testing, and the test atmosphere. [6] p. 7: From Figure 4, the tensile test was taken from a hollow extrusion. If so, please describe the method of extrusion welding in the text. [7] p. 8 Please cite the reference in the Hensel-Spittel constitutive equation. [8] p. 9: Please add an explanation of Lebanov in the text or cite the reference. [9] p. 10-: The figure described in the text does not exist, or the figure number is incorrect. [10] p. 10-: The precipitates are so fine that they cannot be observed with an optical microscope. [11] p. 10-: In the text, alloys are referred to by their No. names and are not described in a unified manner, such as "alloy 1". Such expressions may mislead readers, so please use uniform expressions. [11] Also, the English is too bad. I recommend that you submit your manuscript to an English proofreader.   Best regard,
